# Plasma Antithrombin III Levels Can Be a Prognostic Factor in Liver Cirrhosis Patients with Portal Vein Thrombosis

**DOI:** 10.3390/ijms24097732

**Published:** 2023-04-23

**Authors:** Tsuyoshi Suda, Hajime Takatori, Takehiro Hayashi, Kiichiro Kaji, Kouki Nio, Takeshi Terashima, Tetsuro Shimakami, Kuniaki Arai, Tatsuya Yamashita, Eishiro Mizukoshi, Masao Honda, Kenichiro Okumura, Kazuto Kozaka, Taro Yamashita

**Affiliations:** 1Department of Gastroenterology, Graduate School of Medical Science, Kanazawa University, Kanazawa 920-8641, Japan; 2Department of Radiology, Graduate School of Medical Science, Kanazawa University, Kanazawa 920-8641, Japan

**Keywords:** portal vein thrombosis, cirrhosis, antithrombin III, prognosis, liver failure

## Abstract

Liver function influences the plasma antithrombin (AT)-III levels. AT-III is beneficial for patients with portal vein thrombosis (PVT) and low plasma AT-III levels. However, whether these levels affect prognosis in patients with cirrhosis-associated PVT remains unknown. This retrospective study involved 75 patients with cirrhosis and PVT treated with danaparoid sodium with or without AT-III. The plasma AT-III level was significantly lower in patients with liver failure-related death than in those with hepatocellular carcinoma (HCC)-related death (*p* = 0.005), although the Child–Pugh and albumin-bilirubin (ALBI) scores were not significantly different between these two groups. Receiver operating characteristic curve analysis of the plasma AT-III levels showed cutoff values of 54.0% at 5-year survival. Low plasma AT-III levels (<54.0%) were associated with significantly worse prognosis than high levels in both overall survival (*p* = 0.0013) and survival excluding HCC-related death (*p* < 0.0001). Low plasma AT-III (<54.0%) was also associated with a significantly worse prognosis among patients with Child–Pugh A/B or ALBI grade 1/2 (*p* < 0.0001). Multivariate analyses indicated that low plasma AT-III levels (<54.0%) were an independent prognostic factor for poor survival outcome. Low plasma AT-III levels may be associated with mortality, particularly liver failure-related death, independent of liver function.

## 1. Introduction

Portal vein thrombosis (PVT) is the most common deep venous thrombosis event that occurs in patients with cirrhosis. The overall incidence of PVT is higher in patients with decompensated cirrhosis [1,2] and ranges from 5% at 1 year to 40% at 10 years [3].

The appearance of PVT in patients with cirrhosis may affect hepatic reserve, appearance of ascites, and exacerbation of gastroesophageal varices. In particular, increased mortality has been reported in cases where PVT appears after liver transplantation [4,5,6].

Anticoagulants are often used for the treatment of PVT. Warfarin, heparin products, and direct oral anticoagulants have been utilized, but are associated with an increased risk of bleeding in patients with cirrhosis [7,8].

Antithrombin (AT) III is a coagulation regulator produced by the liver. It exhibits an inhibitory effect on blood coagulation through the action of coagulation enzymes, such as thrombin, plasmin, and IXa, Xa, XIa, and XIIa factors [9]. Regarding the plasma AT levels, at least 70% may be needed to inhibit the coagulation cascade effectively [10]. The AT-III levels have been reported to decrease as a result of impaired liver function [11]. In addition, a prospective study reported that low AT-III levels were significantly associated with PVT in patients with cirrhosis after liver surgery [12] and splenectomy [13].

In a randomized, double-blind controlled trial, treatment with AT-III has proven beneficial for patients with PVT and low AT-III levels. Moreover, the safety of AT-III treatment was documented, and there was no associated risk of bleeding [14]. Recently, a retrospective and multicenter study of AT-III-based therapy for patients with PVT revealed that AT-III-based therapy achieved a favorable response (67.5%) [6]. However, there is no consensus on how AT-III affects prognosis in patients with cirrhosis associated with PVT. Therefore, we retrospectively investigated whether patients with cirrhosis and PVT who received thrombolytic therapy had a different prognosis depending on the AT-III levels.

## 2. Results

### 2.1. Patient Characteristics

Table 1 summarizes the clinical characteristics (*n* = 75). All patients were diagnosed with cirrhosis; 40% had hepatocellular carcinoma (HCC) (*n* = 30), and 70.7% were male (*n* = 53). The median Child–Pugh and ALBI scores were 8 and −1.69, respectively (median modified ALBI grade 2b). The median plasma AT-III level, serum albumin (Alb) level, serum total bilirubin (T-Bil) level, and plasma prothrombin time (PT) activity were 55%, 3.1, 1.2, and 66%, respectively. The median PVT volume was 3.54 mL.

### 2.2. Relationship between Plasma AT-III, PVT Volume, Plasma Fibrinogen Degradation Products/D-Dimer, and Child–PUGH Score

Patients were divided into the Child–Pugh class A, B, and C groups, based on their Child–Pugh score. The plasma AT-III level differed significantly among the three Child–Pugh groups (*p* < 0.0001). The plasma AT-III level was associated with decreasing liver function, in the order of Child–Pugh A to C (Figure 1a).

The volume of PVT did not differ significantly among the three Child–Pugh groups, although the absolute values in the Child–Pugh A and Child–Pugh C groups were smaller than those in the Child–Pugh B group (Figure 1b). Finally, the plasma fibrinogen degradation product (FDP) and D-dimer (DD) levels were the highest in the Child–Pugh C group and increased gradually in the Child–Pugh C group (*p* = 0.0009, *p* = 0.0013, respectively; Figure 1c,d).

### 2.3. Correlations between Plasma AT-III and Clinical Parameters

According to Spearman’s correlation coefficient analysis, plasma AT-III showed significant positive correlations with PT activity, serum Alb, blood platelet concentration, and PVT volume (Figure 2a–d; r = 0.569, *p* < 0.0001; r = 0.454, *p* < 0.0001; r = 0.263, *p* = 0.0226; and r = 0.234, *p* = 0.0431, respectively). The AT-III levels showed negative correlations with Child–Pugh scores, albumin-bilirubin (ALBI) score, serum T-Bil, and the fibrosis-4 (FIB-4) index (Figure 2e–h; r = −0.563, *p* < 0.0001; r = −0.583, *p* < 0.0001; r = −0.523, *p* < 0.0001, and r = −0.379, *p* = 0.0008, respectively). The plasma FDP and DD levels showed no significant intercorrelations.

### 2.4. Plasma AT-III Levels in Liver Failure-Related and HCC-Related Deaths

In total, 45 patients for whom 5-year survival data were available were selected: 30 patients died. The cause of death in these patients was liver failure-related (*n* = 10), HCC-related (*n* = 14), and for the other patients, gastrointestinal bleeding or unknown (*n* = 6).

The plasma AT-III levels, Child–Pugh score, and ALBI score were compared between patients with liver failure-related death and those with HCC-related death. Plasma AT-III levels were significantly lower in patients with liver failure-related death than in those with HCC-related death (*p* = 0.005, Figure 3a). However, the Child–Pugh and ALBI scores were not significantly different between these two groups of patients (Figure 3b,c).

### 2.5. Receiver Operating Characteristic (ROC) Curve Analysis Excluding HCC-Related Death at 5-Year Survival

We performed the ROC curve analysis of 5-year survival for the plasma AT-III levels, Child–Pugh score, and ALBI score. As the prognosis of patients with HCC is poor, patients with HCC-related deaths were excluded in advance [15]. We selected 31 patients for whom survival data at 5 years were available, excluding HCC-related death.

From the ROC curve analysis of plasma AT-III levels, a cutoff value of 54.0% (area under the curve (AUC) 0.892 [95% confidence interval (CI): 0.774–1.01], sensitivity 81.3%, specificity 93.3%) at 5-year survival (Figure 4a) were determined.

The ROC curve analysis for the Child–Pugh and ALBI scores revealed the following: the cutoff values of 8.5 (AUC, 0.827 [95% CI: 0.671–0.984]; sensitivity, 62.5%; specificity, 86.7%) and −1.73 (AUC, 0.858 [95% CI: 0.731–0.986]; sensitivity, 81.3%; specificity, 73.3%) were determined at 5-year survival, respectively (Figure 4a).

We also performed the ROC curve analysis for serum Alb, T-Bil, and plasma PT activity, and compared these results with the respective analysis for the plasma AT-III levels. The ROC curve analysis of serum Alb, T-Bil, and plasma PT activity revealed the following: the cutoff values of 2.85 (AUC 0.815 [95% CI: 0.658–0.972], sensitivity 62.5%, specificity 93.3%), 1.65 (AUC 0.779 [95% CI: 0.608–0.950], sensitivity 68.8%, specificity 80.0%), and 68.0% (AUC 0.773 [95% CI: 0.603–0.943], sensitivity 87.5%, specificity 66.7%) were determined at 5-year survival, respectively (Figure 4b).

### 2.6. Survival According to the Plasma AT-III Levels

We examined whether 5-year survival depended on plasma AT-III levels using Kaplan–Meier survival curve analysis. The cutoff value for this analysis was defined at 54.0%, based on the ROC curve analysis at 5-year survival.

Regarding the overall survival analysis, all patients (*n* = 75) were divided into two groups according to the cutoff value of 54.0%: high plasma AT-III (*n* = 46) and low plasma AT-III (*n* = 29) groups. The analysis revealed that the low plasma AT-III group had a significantly worse prognosis than the high plasma AT-III group (hazard ratio (HR), 3.68; 95% CI, 1.66–8.16; *p* = 0.0013) (Figure 5a).

As previously described, HCC is associated with a poor prognosis. Therefore, we excluded patients with HCC-related deaths and decided to repeat the analysis. Sixty-one patients were selected and divided into the high plasma AT-III (*n* = 36) and low plasma AT-III (*n* = 25) groups, as described above. The characteristics of the patients in these two groups are shown in Appendix A. This analysis also indicated that the low plasma AT-III group was associated with a worse prognosis than the high plasma AT-III group, and this difference was significant (HR, 8.02; 95% CI, 2.84–22.7; *p* < 0.0001) (Figure 5b).

The same analysis was performed based on the Child–Pugh score. The cutoff value for this analysis was defined at nine points, based on the ROC curve analysis at 5-year survival. As expected, patients with a poor Child–Pugh score (*n* = 22) had a worse prognosis than those with a good Child–Pugh score (*n* = 39), and this difference was significant (HR, 8.25; 95% CI, 2.59–26.2; *p* = 0.0004) (Figure 5c).

Survival analysis was also performed based on liver function evaluation. Among patients with a Child–Pugh A/B classification (*n* = 44), those with low plasma AT-III had a significantly worse prognosis than those with high plasma AT-III (HR, 33.5; 95% CI, 5.72–195.9; *p* < 0.0001; Figure 5d). Furthermore, among patients with ALBI grade 1/2 (*n* = 47), patients with low plasma AT-III also had a significantly worse prognosis than patients with high plasma AT-III (HR, 34.1; 95% CI, 6.63–175.2; *p* < 0.0001; Figure 5e). Finally, the numbers of patients with Child–Pugh C (*n* = 17) and ALBI grade 3 (*n* = 14) were small; therefore, no significant differences were observed between the groups with high vs. low plasma AT-III in these settings.

Next, we investigated whether there was a significant difference in survival based on different combinations of the plasma AT-III levels and Child–Pugh scores. Patients with plasma AT-III ≥54% and a Child–Pugh score <9 points were classified into Group 1 (*n* = 28), patients with plasma AT-III ≥54% and a Child–Pugh score ≥9 points or AT-III <54% and a Child–Pugh score <9 points were classified into Group 2 (*n* = 19), and those with plasma AT-III <54% and a Child–Pugh score ≥9 points were classified into Group 3 (*n* = 14). The prognosis worsened from Group 1 and Group 2 to Group 3 in that order, and this difference was significant (*p* < 0.0001). The median survival time was 1463 days and 554 days in Groups 2 and 3, respectively (Figure 5f).

Finally, we performed univariate and multivariate analyses using the Cox proportional hazards model with stepwise Akaike’s Information Criterion. The cutoff value of plasma AT-III was determined as 54%, as described above. According to the ROC curves reported above, the cutoff value of the Child–Pugh and ALBI scores was set at 9 and −1.73 points, respectively. Univariate analysis indicated that low hepatic reserves (based on Child–Pugh and ALBI scores), low plasma AT-III levels, and the presence of HCC were risk factors for poor survival (Table 2). Multivariate analysis indicated that low plasma AT-III levels were independent prognostic factors for poor survival (HR, 5.69; 95% CI, 1.48–21.8; *p* = 0.0011; Table 2). Poor Child–Pugh scores were also considered a prognostic factor with borderline significance (*p* = 0.058; Table 2).

## 3. Discussion

It has been reported that plasma AT-III levels are decreased in patients with cirrhosis, given that liver function is decreased [11]. This finding was also confirmed in our study. However, the volume of PVT was greater in the Child–Pugh B group compared to the Child–Pugh A and Child–Pugh C groups.

This paradoxical observation in patients with cirrhosis could be explained by the disruption of their hemostatic balance, due to loss of both coagulation and anticoagulation factors [16]. An indirect finding in support of this hypothesis is that, when the Child–Pugh score was poor, the plasma FDP and DD levels were high, and the coagulation system may be predominantly fibrinolytic. Therefore, patients with cirrhosis having PVT may have a very complex balance between coagulation and fibrinolysis.

In patients with cirrhosis who have PVT, excluding cases of HCC-related death, ROC curves showed that the plasma AT-III level was a stable prognostic factor at 5 years, with an AUC of 0.892. In terms of sensitivity and specificity, the plasma AT-III levels were >80%. The Child–Pugh and ALBI scores also showed close AUC values, but their sensitivity and specificity were not as good as the plasma AT-III levels.

Kaplan–Meier survival curve analysis revealed that the mortality rate was significantly higher in the group with low plasma AT-III levels at 5-year follow-up. This was also true within the Child–Pugh A/B and ALBI grade 1/2 groups, suggesting that the plasma AT-III levels may be a prognostic factor regardless of hepatic reserve. The group with a high Child–Pugh score or plasma AT-III level had a better prognosis than the group with a poor Child–Pugh score or low plasma AT-III level, and the respective median survival was more than twice as large (1463 vs. 554 days).

In patients undergoing initial liver resection for HCC, AT-III is more specific than indocyanine green for the diagnosis of postoperative liver failure and liver dysfunction [17]. The prognostic impact of plasma AT-III has also been reported in cases other than cirrhosis and PVT. For instance, in patients with end-stage heart failure receiving mechanical circulatory support, a decrease in plasma AT-III level has been associated with an increased risk for acute liver failure [18]. Moreover, in patients with angina pectoris, lower plasma AT-III level was associated with subsequent cardiac events [19]. In addition, in a study of patients with type 2 diabetes, the plasma AT-III level decreased in patients with poor glycemic control, and there was a concern regarding an increased risk of thrombosis in these patients [20].

Conversely, in patients with idiopathic pulmonary fibrosis receiving the tyrosine kinase inhibitor nintedanib, there was a significant inverse correlation between the plasma AT-III level and pulmonary function test parameters, with high baseline plasma AT-III levels being a predictor of poor prognosis. This finding may be explained by the involvement of the coagulation process in pulmonary wound healing and repair [21].

The findings of this study suggest that decreased plasma AT-III levels were associated with liver failure-related death. Interestingly, the results also indicate that patients with decreased plasma AT-III had a poorer prognosis, even among those with similar liver function.

This study has several limitations that should be acknowledged. First, this was a single-center, retrospective study. Second, the patient population had very low liver function, and 40% of patients had coexistent HCC. Third, the number of evaluated patients decreased significantly once the patients with HCC-related death were excluded. Fourth, the criterion for treatment selection, danaparoid sodium with or without AT-III, differed depending on the time period. In addition, maintenance treatment was not mentioned in this study. Fifth, the effect of thrombolytic therapy for PVT on prognosis was not examined.

Although our study was limited to patients with concomitant cirrhosis and PVT, plasma AT-III may be a prognostic factor independent of liver function. Therefore, further studies are needed to elucidate the precise relationship between plasma AT-III levels and liver function. Thus, we consider that it is necessary to examine this issue in patients with chronic liver disease who do not have PVT and HCC.

## 4. Materials and Methods

### 4.1. Study Design and Patients

This retrospective study involved 75 patients with cirrhosis and PVT treated with danaparoid sodium between November 2008 and December 2020.

The treatment protocol has been described previously [22,23]. In brief, PVT was defined as the presence of a contrast defect area in the portal venous system (i.e., superior mesenteric vein or portal vein to intrahepatic portal vein) in the portal venous phase of contrast-enhanced CT. All patients with PVT were treated with danaparoid sodium, with AT-III (combination therapy) or without AT-III (monotherapy), at Kanazawa University Hospital.

Liver function was evaluated using Child–Pugh and albumin-bilirubin (ALBI) scores [24]. As ALBI has a wide coverage of intermediate grades (grade 2), we used the modified ALBI grade, which is a 4-point scale (1, 2a, 2b, and 3) divided into two subgrades with an ALBI score of −2.27 points [25].

The Institutional Review Board (IRB) of Kanazawa University Hospital approved the study treatment strategy and protocol (no. 2016–096). The study was conducted according to the principles of the Declaration of Helsinki. The requirement for obtaining informed consent from the patients was waived by the IRB considering the retrospective nature of the study.

### 4.2. Measurement of PVT

A detailed method of PVT measurement has been described previously [22,23]. In brief, the thrombus volume on portal venous phase images of contrast-enhanced CT (2.5–3-mm slice thickness) was calculated using a three-dimensional image analysis system (Synapse Vincent Ver. 3 and Ver. 5; Fujifilm Medical Co., Tokyo, Japan).

### 4.3. Statistical Analysis

Spearman’s rank-order correlation, ROC curve, Mann–Whitney U test, Kruskal–Wallis test with the post hoc Dunn’s test, chi-squared test, and log-rank test analyses were performed using the GraphPad Prism 7 software package (GraphPad Software, San Diego, CA, USA). The Cox proportional hazard model was constructed using the EZR version 1.53 (Saitama Medical Center, Jichi Medical University, Saitama, Japan) statistical software package, a graphical user interface for R (The R Foundation for Statistical Computing, Vienna, Austria). Statistical significance was defined as a *p*-value < 0.05; *p*-values < 0.05, <0.01, and <0.001 were indicated with *, **, and ***, respectively.

## 5. Conclusions

In patients with cirrhosis with PVT, lower plasma AT-III levels may be associated with mortality, particularly liver failure-related death, independent of liver function.

## Figures and Tables

**Figure 1 ijms-24-07732-f001:**
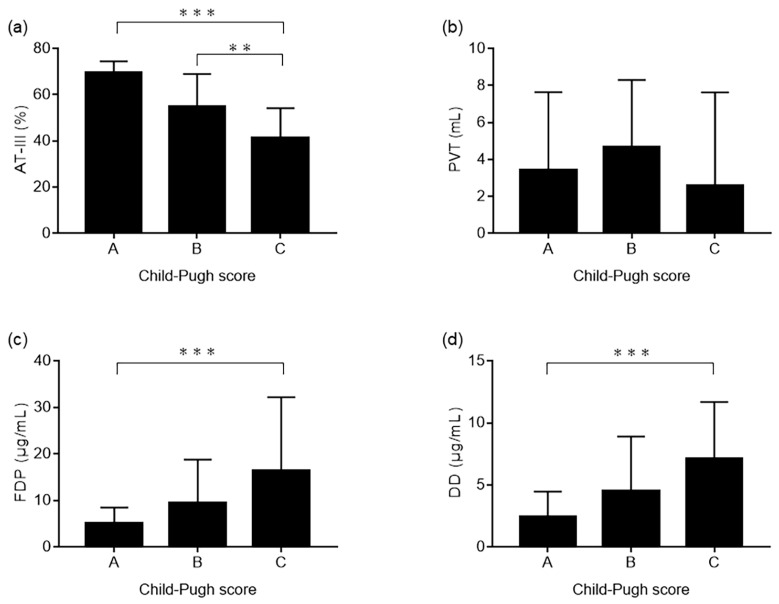
Relationship between plasma AT-III levels, PVT volume, plasma FDP/DD levels, and Child–Pugh score. (**a**) The Kruskal–Wallis test reveals significant differences among Child–Pugh groups A, B, and C (*p* < 0.0001). AT-III level is associated with liver function and decreases gradually from Child–Pugh A to C. (**b**) The volume of PVT is also not significantly different among Child–Pugh A, B, and C groups with the Kruskal–Wallis test. However, the absolute values in the Child–Pugh A and Child–Pugh C groups are smaller than those in the Child–Pugh B group. (**c**,**d**) Finally, the FDP and DD levels are the highest in the Child–Pugh C group and increase gradually to Child–Pugh C (*p* = 0.0009, *p* = 0.0013, respectively). Significant differences in multiple comparisons with the post hoc Dan tests are indicated by asterisks (** *p* < 0.01, *** *p* < 0.001). AT-III, antithrombin III; PVT, portal vein thrombosis; FDP, fibrinogen degradation products; DD, D-dimer.

**Figure 2 ijms-24-07732-f002:**
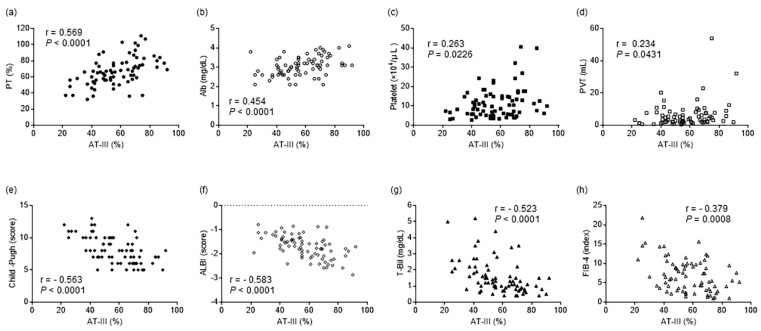
Correlations between the plasma AT-III levels and clinical parameters. (**a**–**d**) Plasma AT-III shows positive correlations with plasma PT activity, serum Alb, blood platelet concentration, and PVT volume. (**e**–**h**) Plasma AT-III shows negative correlations with Child–Pugh score, ALBI score, serum T-Bil, and FIB-4 index. AT-III, antithrombin III; PT, prothrombin time; Alb, albumin; PVT, portal vein thrombosis; ALBI, albumin-bilirubin; T-Bil, total bilirubin; FIB-4, fibrosis-4.

**Figure 3 ijms-24-07732-f003:**
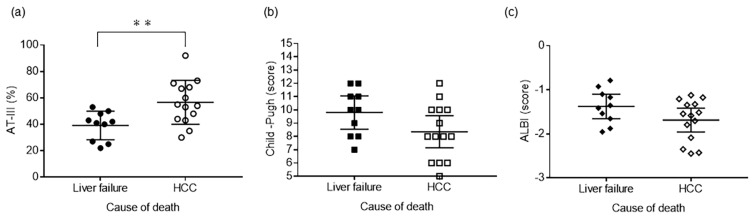
Plasma AT-III levels in liver failure-related and liver cancer-related deaths. (**a**) Plasma AT-III is significantly lower in patients with liver failure-related deaths than in those with HCC-related deaths (*p* = 0.005). (**b**,**c**) The Child–Pugh and ALBI scores do not differ significantly between these two groups of patients. Significant differences are indicated by asterisks (** *p* < 0.01). AT-III, antithrombin III; ALBI, albumin-bilirubin; HCC, hepatocellular carcinoma.

**Figure 4 ijms-24-07732-f004:**
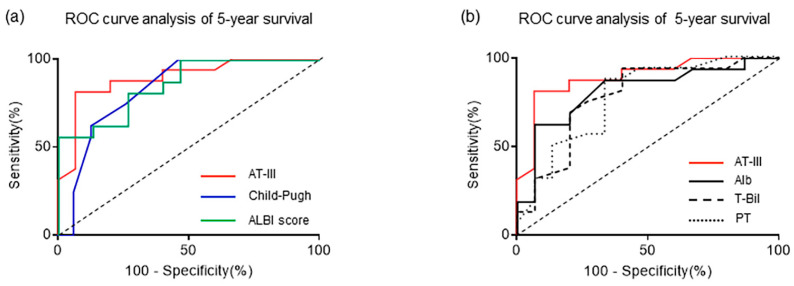
ROC curve analysis excluding HCC-related deaths at 5-year survival. (**a**) The ROC curve analysis for 5-year survival indicates that the cutoff value of the plasma AT-III levels, Child–Pugh score, and ALBI score are 54.0%, 8.5, and −1.73; AUC is 0.892, 0.827, and 0.858; sensitivity is 81.3%, 62.5%, and 81.3%; and specificity is 93.3%, 86.7%, and 73.3%, respectively. (**b**) The ROC curve analysis for 5-year survival showing the cutoff value of serum Alb, T-Bil, and plasma PT activity to be 2.85, 1.65, and 68.0%; AUC to be 0.815, 0.779, and 0.773; sensitivity to be 62.5%, 68.8%, and 87.5%; and specificity to be 93.3%, 80.0%, and 66.7%, respectively. ROC, receiver operating characteristic; HCC, hepatocellular carcinoma; AT-III, antithrombin III; ALBI, albumin-bilirubin; AUC, area under the ROC curve; Alb, albumin; T-Bil, total bilirubin; PT, prothrombin time.

**Figure 5 ijms-24-07732-f005:**
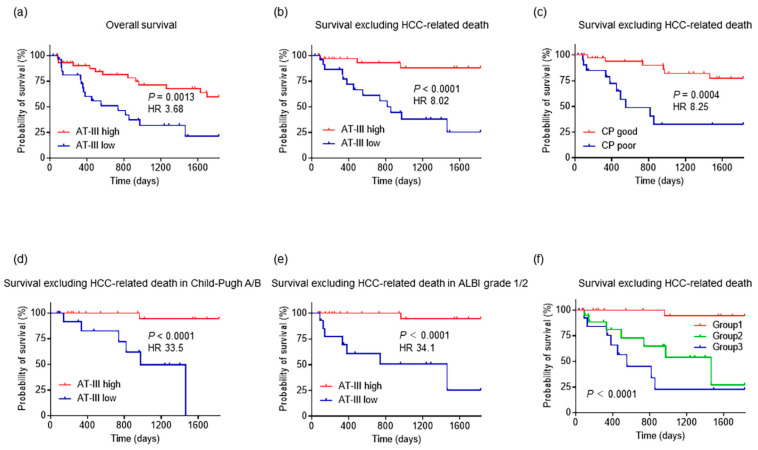
Survival analysis according to plasma AT-III levels at 5-year survival. (**a**) Kaplan–Meier survival analysis indicating a significant correlation of low plasma AT-III levels (<54%) with a worse prognosis in overall survival (*p* = 0.0013). (**b**) Regarding survival excluding HCC-related death, the group with low plasma AT-III levels (<54%) also shows a poor prognosis (*p* < 0.0001). (**c**) Poor Child–Pugh score (≥9 points) indicates a significant correlation with a poor prognosis in survival, excluding HCC-related deaths (*p* = 0.0004). (**d**) Regarding patients with Child–Pugh A/B, the group with low plasma AT-III levels (<54%) shows a poor prognosis in survival, excluding HCC-related death (*p* < 0.0001). (**e**) In ALBI grade 1/2 patients, the group with low plasma AT-III levels (<54%) also indicates a poor prognosis in survival, excluding HCC-related deaths (*p* < 0.0001). (**f**) Patients are divided into three groups: Group 1, with high plasma AT-III levels (≥54%) and a good Child–Pugh score (<9 points), Group 2, with high plasma AT-III levels (≥54%) or a good Child–Pugh score (<9 points), and Group 3, with low AT-III levels (<54%) and a poor Child–Pugh score (≥9 points). According to Kaplan–Meier survival curve analysis, the prognosis worsens from Group 1 to Group 2 to Group 3 in that order, and this difference is significant (*p* < 0.0001). AT-III, antithrombin III; HCC, hepatocellular carcinoma; ALBI, albumin-bilirubin.

**Table 1 ijms-24-07732-t001:** Patients’ characteristics. Data are presented as medians (interquartile ranges) unless specified otherwise.

Characteristics	Patients (*n* = 75)
Age (years)	67 (60–73)
Sex, male, *n* (%)	53 (70.7)
Etiology, HCV/HBV/others, *n* (%)	35 (46.7)/11 (14.7)/29 (38.7)
HCC present, *n* (%)	30 (40.0)
PVT (mL)	3.54 (1.91–7.7)
Child–Pugh score	8 (6–10)
ALBI score	−1.69 (−2.08–−1.38)
Modified ALBI grade, 1/2a/2b/3, *n* (%)	1 (1.3)/9 (12.0)/46 (61.3)/19 (25.3)
FIB-4 index	6.13 (3.42–9.74)
AT-III (%)	55 (45–70)
Platelets (×10⁴/μL)	9.7 (5.7–14.8)
Alb (mg/dL)	3.1 (2.7–3.4)
T-Bil (mg/dL)	1.2 (0.8–2.0)
PT activity (%)	66 (56–77)
FDP (μg/mL)	9.1 (4.6–17.8)
DD (μg/mL)	4.5 (2.2–8.8)

HCV, hepatitis C virus; HBV, hepatitis B virus; HCC, hepatocellular carcinoma; PVT, portal vein thrombosis; ALBI, albumin-bilirubin; FIB-4, fibrosis-4; AT-III, antithrombin III; Alb, albumin; T-Bil, total bilirubin; PT, prothrombin time; FDP, fibrinogen degradation products; DD, D-dimer.

**Table 2 ijms-24-07732-t002:** Multivariate analysis of clinical characteristics for survival, excluding HCC-related death, using the Cox proportional hazards model with stepwise AIC. Significant differences are indicated by asterisks (* *p* < 0.05, ** *p* < 0.01, *** *p* < 0.001). HCC, hepatocellular carcinoma; AIC, Akaike’s Information Criterion; HR, hazard ratio; CI, confidence interval; n.s., not significant; HCV, hepatitis C virus; HBV, hepatitis B virus; ALBI, albumin-bilirubin; AT-III, antithrombin III.

	Univariate	HR (95% CI)	Multivariate	HR (95% CI)
Age	n.s.	—		
Sex	n.s.	—		
Etiology (HCV/HBV/others)	n.s.	—		
Child–Pugh score (≥9)	** 0.0013	5.52 (1.95–15.6)	0.058	2.89 (0.96–8.65)
ALBI (>−1.73)	* 0.011	5.14 (1.46–18.2)	n.s	—
AT-III (<54%)	*** 0.0009	8.56 (2.40–30.5)	* 0.011	5.69 (1.48–21.8)
HCC (present)	* 0.045	2.73 (1.02–7.30)	n.s	—

## Data Availability

The data that support the findings of this study are available on request from the corresponding author. The data are not publicly available due to privacy or ethical restrictions.

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
