# Peer review of "Plasma Antithrombin III Levels Can Be a Prognostic Factor in Liver Cirrhosis Patients with Portal Vein Thrombosis"

_ijms, 2023, doi:10.3390/ijms24097732_

Round 1
Reviewer 1 Report
Dear Editor,
I have read with great interest the manuscript entitled "Plasma antithrombin III level can be a prognostic factor in liver cirrhosis patients with portal vein thrombosis" submitted by Suda et al. to IJMS. However, there are some issues that need to be addressed before further processing.
Introduction: too short. The authors should include more data on the physiopathology of ATIII and the impact of liver disease. Describe the importance of this study and specify the aims of the study.
Results: The figure description (especially for figure 4) are too long.
Materials and Methods: what test was used to study the distribution of data? How were model built? How were variables selected for multivariate analysis? Was p-value two tailed?
Author Response
Reviewer 1
The authors would like to thank the reviewer for their constructive critique to improve the manuscript. We have made every effort to address the issues raised and to respond to all comments. The revisions are indicated in red font in the revised manuscript. Please, find next a detailed, point-by-point response to the reviewer's comments. We hope that our revisions will meet the reviewer’s expectations.
Response
We would like to thank the reviewers for their constructive critique to improve the manuscript. We have made every effort to address the issues raised and to respond to all comments. The revisions are indicated in red font in the revised manuscript. Please find below a detailed, point-by-point response to the reviewer’s comments. We hope that our revisions will meet the reviewer’s expectations.
After receiving the reviewer’s comment, we reexamined our manuscript and found some issues with the data and analyses.
We apologize for the delay in our response due to the time needed to recheck the data and perform the necessary re-analyses. The results of the re-analysis revealed incorrect values for some data, which have been corrected. The corrections are presented in red font.
For example, we initially stated that the plasma AT-III levels did not differ significantly among the three Child–Pugh groups, although Child–Pugh group B had the highest value. However, the re-analysis revealed that the plasma AT-III level was significantly different among the three groups and decreased in the order of Child–Pugh group A to C.
Please refer to the revised manuscript for other minor changes. Fortunately, we consider that no major changes to the overall content were necessary.
- Introduction: too short. The authors should include more data on the physiopathology of ATIII and the impact of liver disease. Describe the importance of this study and specify the aims of the study.
Response
We would like to thank the reviewer for this valuable comment. We have added more information in the Introduction concerning the importance of AT-III in the treatment of patients with cirrhosis with PVT, the prognostic significance of the presence of PVT, the general in vivo anticoagulant effect of AT-III, and the therapeutic effect of AT-III in the treatment of patients with cirrhosis with PVT.
- Results: The figure description (especially for figure 4) are too long.
Response
We agree with the reviewer’s observation that the description for Figure 4 was too long. Thus, we have retained only the explanation of the ROC curve analysis excluding HCC-related death at 5 years for AT-III, Child-Pugh, ALBI scores, serum Alb, T-Bil, and plasma PT activity.
We have also simplified the explanations of the other figure legends. Please check them.
- Materials and Methods: what test was used to study the distribution of data? How were model built? How were variables selected for multivariate analysis? Was p-value two tailed?
Response
We would like to thank the reviewer for the questions. Although tests of normality using histograms and statistical analysis such as the D'Agostino-Pearson omnibus test or Shapiro–Wilk test were also performed, all data were analyzed using non-parametric tests because of the small number of patients in this study.
Unfortunately, we do not have the statistical expertise to state what models were used by the statistical software, but we used the GraphPad Prism 7 software package and the EZR to perform the statistical analysis, as per the manufacturer’s instructions.
Regarding the factors used in the multivariate analysis using stepwise AIC, all factors, such as age, sex, etiology (HCV/HBV/others), Child–Pugh score (<9/≥9), ALBI (<-1.73/≥-1.73), AT-III (≥54%/<54%), and HCC (absent/present) were allowed.
The results remained the same, if only factors identified as significant in the univariate analysis, such as the Child–Pugh score (<9/≥9), ALBI (<-1.73/≥-1.73), AT-III (≥54%/<54%), and HCC (absent/present), were included.
Two-tailed test was performed for P-values.

Reviewer 2 Report
The present study entitled “Plasma antithrombin III level can be a prognostic factor in liver 2 cirrhosis patients with portal vein thrombosis” aims to evaluate whether liver cirrhosis patients with PVT who received thrombolytic therapy present a differential prognosis based on AT-III levels. The results showed that in cirrhotics with PVT the poor Child–Pugh scores did not correlate with the lower plasma AT-III levels. Moreover, lower plasma AT-III levels were associated with mortality, particularly liver failure-related death, independent of liver function.
This work is interesting. However, there are major issues that should be addressed.
The major limitation of the current study is the relatively small sample size. In fact, only 45 patients with cirrhosis and without HCC were included in the current study. This fact significantly restricts the quality of the extracted conclusions.
The single institutional retrospective nature of the study is one limitation as well, as already mentioned in the current study.
Another concern is the time range of patients’ recruitment. From 2008 to 2020 patients’ management may vary, leading to potential bias.
ALBI abbreviation should be defined at the first presence in the manuscript (it is defined in the materials and methods section)
The figure legends should briefly summarize the content of each figure, not to repeat the results. Please revise.
Author Response
Reviewer 2
The present study entitled “Plasma antithrombin III level can be a prognostic factor in liver 2 cirrhosis patients with portal vein thrombosis” aims to evaluate whether liver cirrhosis patients with PVT who received thrombolytic therapy present a differential prognosis based on AT-III levels. The results showed that in cirrhotics with PVT the poor Child–Pugh scores did not correlate with the lower plasma AT-III levels. Moreover, lower plasma AT-III levels were associated with mortality, particularly liver failure-related death, independent of liver function.
This work is interesting. However, there are major issues that should be addressed.
Response
We would like to thank the reviewers for their constructive critique to improve the manuscript. We have made every effort to address the issues raised and to respond to all comments. The revisions are indicated in red font in the revised manuscript. Please find below a detailed, point-by-point response to the reviewer’s comments. We hope that our revisions will meet the reviewer’s expectations.
After receiving the reviewer’s comment, we reexamined our manuscript and found some issues with the data and analyses.
We apologize for the delay in our response due to the time needed to recheck the data and perform the necessary re-analyses. The results of the re-analysis revealed incorrect values for some data, which have been corrected. The corrections are presented in red font.
For example, we initially stated that the plasma AT-III levels did not differ significantly among the three Child–Pugh groups, although Child–Pugh group B had the highest value. However, the re-analysis revealed that the plasma AT-III level was significantly different among the three groups and decreased in the order of Child–Pugh group A to C.
Please refer to the revised manuscript for other minor changes. Fortunately, we consider that no major changes to the overall content were necessary.
- The major limitation of the current study is the relatively small sample size. In fact, only 45 patients with cirrhosis and without HCC were included in the current study. This fact significantly restricts the quality of the extracted conclusions.
Response
We agree with the reviewer’s comment. We aggressively treat the patients if they have HCC with anticoagulant when we find PVT. This is because PVT has already been reported to cause further deterioration of hepatic reserve, and patients with HCC may be at a disadvantage if their hepatic reserve is compromised, resulting in limiting their treatment options. Therefore, the number of patients with HCC who are anticoagulated is higher. Another reason is the fact that PVT is more likely to occur in patients with advanced cirrhosis, but HCC is similarly more likely to occur in such patients. As the reviewer point out, we hope that similar investigations for patients with chronic liver disease without HCC will be necessary in the future.
We have discussed this issue as a limitation in the revised manuscript as follows:
“Second, the patient population had a very low liver function and 40% of patients had coexistent HCC. Third, the number of evaluated patients decreased significantly once the patients with HCC-related death were excluded.” (Lines 276–277)
“Therefore, further studies are needed to elucidate the precise relationship between plasma AT-III levels and liver function. Thus, we consider that it is necessary to examine this issue in patients with chronic liver disease who do not have PVT and HCC.” (Lines 284-287)
- The single institutional retrospective nature of the study is one limitation as well, as already mentioned in the current study.
Response
We agree with the reviewer. We have discussed this issue as a limitation in the revised manuscript as follows:
“First, this was a single-center, retrospective study.” (Lines 275–276)
- Another concern is the time range of patients’ recruitment. From 2008 to 2020 patients’ management may vary, leading to potential bias.
Response
First, all patients with PVT were treated with danaparoid sodium with AT-III (combination therapy) or without AT-III (monotherapy). However, the criteria for treatment selection differ, depending on the time period. In addition, maintenance treatment is not mentioned at this time. Therefore, we believe that this could also be a potential bias. We have discussed this limitation in the revised manuscript as follows:
“Fourth, the criterion for treatment selection, danaparoid sodium with or without AT-III, differed depending on the time period. In addition, maintenance treatment was not mentioned in this study. Fifth, the effect of thrombolytic therapy for PVT on prognosis was not examined.” (Lines 279–282)
- ALBI abbreviation should be defined at the first presence in the manuscript (it is defined in the materials and methods section)
Response
We have made this correction, as per the reviewer’s suggestion.
- The figure legends should briefly summarize the content of each figure, not to repeat the results. Please revise.
Response
We agree with the reviewer’s observation that the description for Figure 4 was too long. Thus, we have retained only the explanation of the ROC curve analysis excluding HCC-related death at 5 years for AT-III, Child-Pugh, ALBI scores, serum Alb, T-Bil, and plasma PT activity.
We have also simplified the explanations of other Figure legends. Please check them.

Reviewer 3 Report
I would like to send compliments to the authors for the originality of the study.
Comments:
1. Please elaborate more about Antithrombin (AT) III and its physiological function in organism, so that the reader could understand it's pathophysiology.
2. In Figure 1 and three you don't have to write p values or ns. It would be much better if the authors would draw the line between columns where there is significant difference between groups and mark above the line with a star sign (*). So if the difference is between groups p<0.050 you can use one *, if it is less than 0.010 you can use two stars, and if p values is 0.001 or less you can use three ***.
3. Please clarify in methodology section what post hoc tests did you use.
4. Please elaborate more about limitations and strenghts of the study.
5. Please, elaborate in cocnlusion possible significance of this study for clinicians.
Author Response
Reviewer 3
I would like to send compliments to the authors for the originality of the study.
Comments:
Response
We would like to thank the reviewers for their constructive critique to improve the manuscript. We have made every effort to address the issues raised and to respond to all comments. The revisions are indicated in red font in the revised manuscript. Please find below a detailed, point-by-point response to the reviewer’s comments. We hope that our revisions will meet the reviewer’s expectations.
After receiving the reviewer’s comment, we reexamined our manuscript and found some issues with the data and analyses.
We apologize for the delay in our response due to the time needed to recheck the data and perform the necessary re-analyses. The results of the re-analysis revealed incorrect values for some data, which have been corrected. The corrections are presented in red font.
For example, we initially stated that the plasma AT-III levels did not differ significantly among the three Child–Pugh groups, although Child–Pugh group B had the highest value. However, the re-analysis revealed that the plasma AT-III level was significantly different among the three groups and decreased in the order of Child–Pugh group A to C.
Please refer to the revised manuscript for other minor changes. Fortunately, we consider that no major changes to the overall content were necessary.
- Please elaborate more about Antithrombin (AT) III and its physiological function in organism, so that the reader could understand it's pathophysiology.
Response
We would like to thank the reviewer for this valuable comment. We have added more information in the Introduction concerning the importance of AT-III in the treatment of patients with cirrhosis with PVT, the prognostic significance of the presence of PVT, the general in vivo anticoagulant effect of AT-III, and the therapeutic effect of AT-III in the treatment of patients with cirrhosis with PVT.
- In Figure 1 and three you don't have to write p values or ns. It would be much better if the authors would draw the line between columns where there is significant difference between groups and mark above the line with a star sign (*). So if the difference is between groups p<0.050 you can use one *, if it is less than 0.010 you can use two stars, and if p values is 0.001 or less you can use three ***.
Response
We would like to thank the reviewer for these suggestions. We have revised the manuscript following your suggestions. Please refer to the revised manuscript.
- Please clarify in methodology section what post hoc tests did you use.
Response
We have made this correction, as per the reviewer’s suggestion.
Please refer to the revised manuscript.
- Please elaborate more about limitations and strenghts of the study.
Response
We agree with the reviewer’s comment.
We treat patients aggressively with anticoagulant if they have HCC when we find PVT. This is because PVT has already been reported to cause further deterioration of hepatic reserve, and patients with HCC may be at a disadvantage if their hepatic reserve is compromised, resulting in limiting their treatment options. Therefore, the number of patients with HCC who are anticoagulated is higher. Another reason is the fact that PVT is more likely to occur in patients with advanced cirrhosis, but HCC is similarly more likely to occur in such patients. As the reviewer point out, we also agree that this is a major limitation of this study.
In addition, although all patients with PVT were treated with danaparoid sodium with AT-III (combination therapy) or without AT-III (monotherapy), the criteria for treatment selection differ, depending on the time period. In addition, maintenance treatment is not mentioned at this time. Therefore, we believe that this could also be a potential bias.
We have discussed this issue as a limitation in the revised manuscript as follows:
“Second, the patient population had very low liver function and 40% of patients had coexistent HCC. Third, the number of evaluated patients decreased significantly once the patients with HCC-related death were excluded.” (Lines 276–277)
“Fourth, the criterion for treatment selection, danaparoid sodium with or without AT-III, differed depending on the time period. In addition, maintenance treatment was not mentioned in this study. Fifth, the effect of thrombolytic therapy for PVT on prognosis was not examined.” (Lines 279–282)
- Please, elaborate in cocnlusion possible significance of this study for clinicians.
Response
The results of this study present the possibility that low plasma AT-III levels may be associated with mortality, particularly liver failure-related death, independent of liver function.
In particular, AT-III may be an independent prognostic factor different from Child–Pugh score and ALBI grade.
Therefore, we believe that measuring plasma AT-III will help predict the patient’s prognosis.
However, this study was performed in patients with PVT and a high number of HCC complications. Thus, we consider that further research is needed.
We have discussed this issue as a limitation in the revised manuscript as follows:
“Therefore, further studies are needed to elucidate the precise relationship between plasma AT-III levels and liver function. Thus, we consider that it is necessary to examine this issue in patients with chronic liver disease who do not have PVT and HCC.” (Lines 284-287)

Round 2
Reviewer 1 Report
The authors have addressed the reviewers' requests. I believe the manuscript is now suitable for publication.
Reviewer 2 Report
The authors had adequately replied my comments and revised the manuscript. I think this manuscript in the present form can be considered for publication in IJMS.